# Estimating direct and indirect genetic effects on offspring phenotypes using genome-wide summary results data

Nicole M. Warrington [1,2,3,4,5]✉, Liang-Dar Hwang [1,2], Michel G. Nivard [6] & David M. Evans [1,2,3]

Estimation of direct and indirect (i.e. parental and/or sibling) genetic effects on phenotypes is becoming increasingly important. We compare several multivariate methods that utilize summary results statistics from genome-wide association studies to determine how well they estimate direct and indirect genetic effects. Using data from the UK Biobank, we contrast point estimates and standard errors at individual loci compared to those obtained using individual level data. We show that Genomic structural equation modelling (SEM) outperforms the other methods in accurately estimating conditional genetic effects and their standard errors. We apply Genomic SEM to fertility data in the UK Biobank and partition the genetic effect into female and male fertility and a sibling specific effect. We identify a novel locus for fertility and genetic correlations between fertility and educational attainment, risk taking behaviour, autism and subjective well-being. We recommend Genomic SEM be used to partition genetic effects into direct and indirect components when using summary results from genome-wide association studies.

[1] Institute for Molecular Biosciences, The University of Queensland, St Lucia, QLD, Australia. [2] The University of Queensland Diamantina Institute, The University of Queensland, Woolloongabba, QLD, Australia. [3] Medical Research Council Integrative Epidemiology Unit at the University of Bristol, Bristol, UK. [4] Population Health Sciences, Bristol Medical School, University of Bristol, Bristol, UK. [5] Faculty of Medicine and Health Sciences, Norwegian University of Science and Technology, Trondheim, Norway. [6] Department of Biological Psychology, Faculty of Behaviour and Movement Sciences, VU University, Amsterdam, The Netherlands. ✉email: n.warrington@uq.edu.au

There is growing interest in estimating maternal genetic effects, via the intrauterine environment, on offspring outcomes (for example refs. [1–3]) and also in elucidating the causal effect of maternal environmental exposures on offspring outcomes (for example refs. [3–5]). Likewise, it is also important to estimate the direct effect of an individual's own genotype on their own phenotype independent of any indirect parental effects. These estimates can subsequently be used in Mendelian randomization studies to make causal inferences, without introducing biases due to dynastic effects and assortative mating[6–8]. However, for correct inference to be made regarding maternal genetic effects and an individual's own genetic effect independent of parental effects, analyses must adjust for the individual's own genetic effect on their own outcome as well as one or both parents. This adjustment has traditionally been performed using conditional analyses applied to individual level genotypes from mother-offspring pairs or parent-offspring trios; however, there are few cohorts worldwide with large numbers of genotyped mother-offspring pairs or parent-offspring trios with offspring phenotypes, leading to limited statistical power in many studies. We have developed a structural equation model (SEM) that can partition genetic effects into maternal and offspring mediated components[9]. This model can incorporate data from genotyped mother-offspring pairs with offspring phenotypes, mother-offspring pairs with maternal genotypes and both mother and offspring phenotypes, individuals with their own genotype and phenotype and mothers with their own genotype and their offspring's phenotype. We have recently described how the maternal and offspring partitioning from this SEM can be used to facilitate large-scale two-sample Mendelian randomization studies investigating whether maternal exposures are causally related to offspring outcomes[10].

Although our SEM is flexible in terms of incorporating many study designs, it is computationally intensive when using individual level data, prohibiting its use for genome-wide association studies (GWAS). Therefore, we wanted to identify other existing methods that could be used on summary results statistics from GWAS to estimate the conditional maternal (paternal) and offspring genetic effects on a trait. There are a number of multivariate methods available that utilize summary statistics from GWAS of multiple traits. For example, metaCCA[11], metaUSAT[12], MTAG[13], TATES[14], $S_{HOM}$ and $S_{HET}$[15], mtCOJO[16] and most recently Genomic SEM[17] are a subset of the multivariate methods that have been proposed for use with GWAS summary statistics to increase statistical power to detect an association with a correlated set of traits and diseases. Although we are interested in combining the summary statistics of the same trait from different genotypes (i.e. the individuals' own genotype and their mother's genotype), we hypothesize that some of these methods could be appropriate. However, because we are interested in using the adjusted maternal and offspring genetic effect in downstream analyses, such as Mendelian randomization, we need methods that would provide unbiased estimates of parental and offspring genetic effects (and standard errors) for each variant. In addition, if we are to use publicly available summary statistics from large GWAS, such a method would need to account for any known or unknown overlap of individuals contributing to maternal (paternal) and offspring GWAS.

In this manuscript, we compare several different multivariate methods to identify the most appropriate method for partitioning the genetic effect of a trait into maternal and offspring components, based on how well the effect estimates compare to those from our SEM using individual level data, the computational time and how well the method accounts for unknown sample overlap. We use birth weight to compare the different methods as we have a substantial number of known associated genetic loci for birth weight, with the genetic effect partitioned into maternal and offspring genetic components. We subsequently use the most appropriate method to conduct conditional GWAS of fertility, partitioning the effects into parental and offspring mediated components providing evidence for how these different loci exert their effect on number of children in a family.

## Results

We searched the literature for multivariate methods that fit the following four criteria: (1) had published code or software, (2) used summary results statistics and did not require individual level data, (3) accounted for sample overlap and (4) produced an effect size estimate and standard error for each trait. In addition to our published structural equation model (SEM)[9] (which can use either individual level data, or variance-covariance matrices, which can be constructed using GWAS summary results statistics) and linear approximation of the SEM[3], we identified three published methods including multi-trait analysis of GWAS (MTAG)[13], multi-trait-based conditional and joint analysis using GWAS summary data (mtCOJO)[16] and Genomic SEM[17]. MTAG is a multivariate method, which uses genome-wide GWAS summary results from multiple correlated phenotypes to increase power to detect pleiotropic loci. mtCOJO is another multivariate method, which uses summary results data but is designed to estimate genetic effects on a trait conditional on a correlated phenotype(s). Although MTAG and mtCOJO are not specifically designed to partition genetic effects into maternal and offspring components (i.e. by conditioning on a correlated genotype), they are user friendly and computationally efficient, and given the dearth of existing software packages to generate conditional genetic effect estimates using genome-wide summary results data, we were interested in investigating whether they would approximate the effects of interest accurately. Genomic SEM on the other hand is a highly flexible (albeit computationally intensive) method that allows users to specify a wide range of models to fit to the data. A summary of each of the methods and their underlying assumptions is provided in Table 1.

**Birth weight GWAS**. Approximately 19 million genetic variants were included in the GWAS analysis of own and offspring birth weight that passed our filtering criteria (INFO score <0.4 and minor allele frequency <0.1%). We excluded variants from the SEM using summary statistics if the minor allele frequency in the sample was <0.5%; this led to ~11 million genetic variants with results. MTAG also implements additional filtering criteria; variants with missing values, variants that are not SNPs, variants with duplicated rs numbers and variants that are strand ambiguous are excluded leading to ~14 million genetic variants with results.

We used bivariate LD score regression[18] to estimate the sample overlap between the GWAS of own and offspring birth weight. We observed a regression intercept of 0.1287 (0.0078) in the analyses where there were individuals in both the GWAS of own and offspring birth weight, indicating that ~91,790 individuals were in both GWAS (true overlap is 85,503 individuals). In the analyses where there were unique individuals in the GWAS of own and offspring birth weight, the observed regression intercept from LD score regression was 0.0161 (0.0064), indicating that ~8396 individuals were in both GWAS (true overlap is 0 individuals). These estimates of sample overlap were used in the SEM analysis using covariance matrices derived from the GWAS summary statistics.

**Comparison with SEM using individual level data**. We compared the effect size and standard errors estimated using the SEM with the individual level data to those estimated using methods

**Table 1 Summary of each of the methods used to derive maternal and offspring specific genetic effects using summary statistics from a GWAS of own birth weight and a GWAS of offspring birth weight.**

| Method | Major assumptions | Data used | Variant exclusions |
|---|---|---|---|
| SEM using summary statistics | • Multivariate normal outcomes<br>• Allele frequency, beta coefficient and SNP variance are consistent across the three groups (individuals with own birth weight only, with offspring birth weight only or with both own and offspring birth weight)<br>• LD reference sample and GWAS samples all drawn from the same population for LD score regression analysis<br>• The effect sizes for each genotype are identically normally distributed with mean zero and the same variance for LD score regression analysis | • Covariance matrices derived from:<br>• GWAS summary results data for own and offspring birth weight<br>• Estimated sample overlap using bivariate LD score regression with a phenotypic correlation between own and offspring birth weight of 0.24<br>• European reference panel from LD score regression | Minor allele frequency <0.5% |
| Linear approximation of SEM | • No sample overlap between GWAS of offspring birth weight and GWAS of own birth weight | • GWAS summary results data for own and offspring birth weight | None |
| MTAG | • LD reference sample and GWAS samples all drawn from the same population for LD score regression analysis<br>• The effect sizes for each genotype are identically normally distributed with mean zero and the same variance for LD score regression analysis<br>• All SNPs share the same variance-covariance matrix of effect sizes across traits | • GWAS summary results data for own and offspring birth weight<br>• European reference panel from LD score regression | Variants with missing values, that are not SNPs, with duplicated rs numbers or that are strand ambiguous |
| mtCOJO | • LD reference sample and GWAS samples all drawn from the same population for LD score regression analysis<br>• The effect sizes for each genotype are identically normally distributed with mean zero and the same variance for LD score regression analysis | • GWAS summary results data for own and offspring birth weight<br>• Reference panel of 50,000 randomly sampled individuals from the UK Biobank<br>• European reference panel from LD score regression | Multi-allelic variants |
| Genomic SEM | • LD reference sample and GWAS samples all drawn from the same population for LD score regression analysis<br>• The effect sizes for each genotype are identically normally distributed with mean zero and the same variance for LD score regression analysis<br>• All SNPs share the same variance-covariance matrix of effect sizes across traits | • European reference panel from LD score regression<br>• GWAS summary results data for own and offspring birth weight | None |

based on the summary statistics from the GWAS of own and offspring birth weight for the 300 autosomal genome-wide significant SNPs identified in the latest GWAS of birth weight[3]. Due to the additional exclusions, MTAG had 258 SNPs available for comparison and the SEM using summary statistics had 298 SNPs. We use the estimates from the SEM using individual level data as a baseline comparator as we have previously shown that they are asymptotically unbiased estimates of the maternal and offspring genetic effects[9]. Figure 1 (also summarized in Table 2) indicates that the effect sizes for each of the 300 genetic variants are accurately estimated using the SEM based on covariance matrices derived from the summary statistics, the linear approximation of the SEM or Genomic SEM, and these do not appear to be influenced by sample overlap. Given mtCOJO and MTAG were not developed to estimate maternal and offspring specific genetic

effects, it is perhaps not surprising that there appears to be a slight underestimation of the effect sizes using mtCOJO, which is consistent with and without sample overlap. In contrast, the effect size estimates from MTAG (which also is not explicitly developed for estimating maternal and offspring genetic effects) differ from the SEM effect sizes for both the maternal and offspring effect, with and without sample overlap.

The comparison of the standard errors for the genetic effects for each of the 300 genetic variants are displayed in Fig. 2 and a summary of the comparison is presented in Table 2. The standard errors for both the maternal and offspring effect are comparable to the SEM using individual level data when using Genomic SEM, both with and without sample overlap. They are also comparable using the linear approximation of the SEM when there is no sample overlap, but are slightly inflated relative to the SEM using

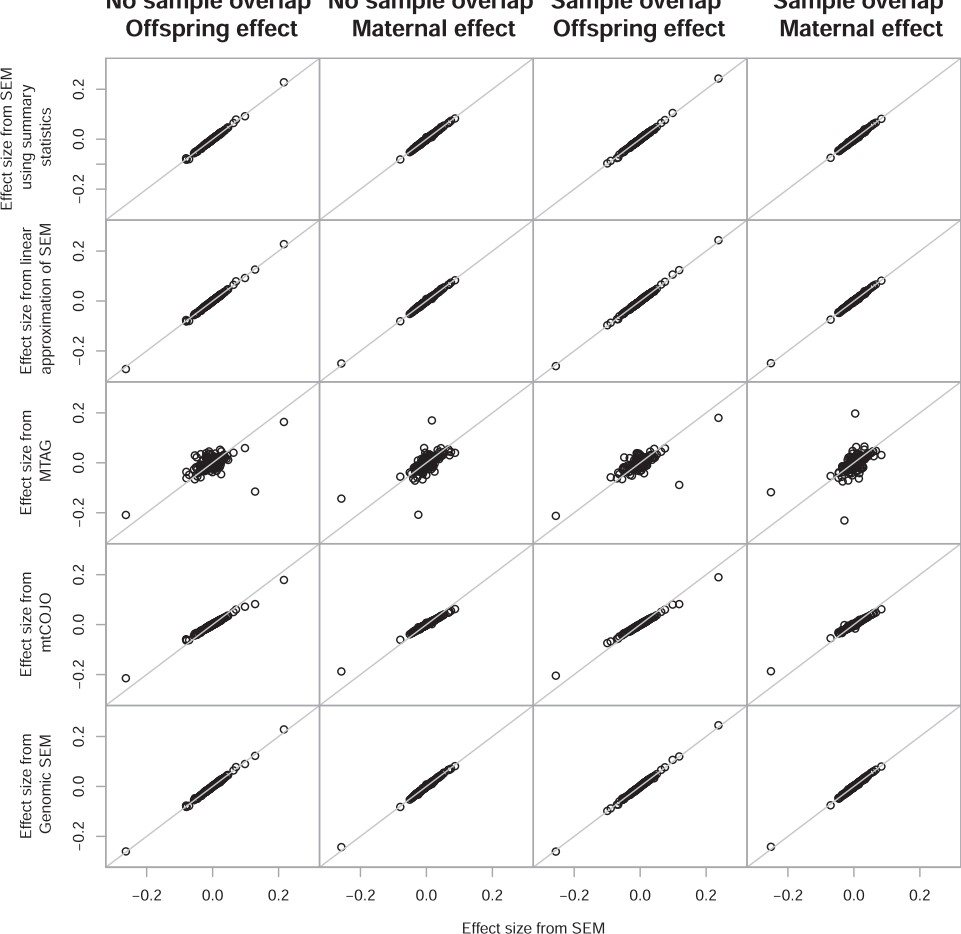

**Fig. 1 Comparison of the effect size estimates from the SEM using individual level data (*x*-axis) and the various different methods using the summary statistics from the GWAS of own and offspring birth weight (*y*-axis) for the 300 autosomal genome-wide significant SNPs from Warrington et al.[3].** The columns summarize the results from the analysis including unique individuals in the GWAS of own and offspring birth weight for the offspring and maternal effect, respectively, followed by the results from the analysis where there were overlapping samples in the GWAS of own and offspring birth weight for the offspring and maternal effect.

individual level data when there is sample overlap that is not accounted for. This is expected as the standard error equations for the linear approximation would need to adjust for twice the covariance between estimates of the maternal and offspring genetic effects when there is sample overlap. In contrast, the standard errors for the maternal and offspring effects are accurately estimated using the SEM based on covariance matrices derived from the summary statistics when there is sample overlap that has been estimated using LD score regression and incorporated in the model, but they are underestimated when there is no sample overlap. This could be due to the small sample overlap that was estimated by LD score regression (8396 individuals were estimated to overlap both GWAS when in reality there were no individuals overlapping. This could be due to e.g. LD score regression identifying cryptic relatedness across the GWAS) and included in the SEM using covariance matrices. We showed in the initial paper describing the SEM[9] that there is an increase in power, due to a reduction in the standard error, when individuals with both their own and their offspring's phenotype are included in a model that specifies this relationship. Therefore, the 8396 individuals estimated to overlap between the GWAS will result in a reduction of the standard error in the SEM using summary statistics (where we model this relationship) in comparison to the SEM using individual level data (where no sample overlap is modelled). As expected due to the difference in

purpose of the method, the standard errors estimated using MTAG and mtCOJO were smaller than those estimated by the SEM using individual level data for the maternal and offspring genetic effect, with and without sample overlap, with the largest difference for MTAG.

We conducted heterogeneity tests for the 300 autosomal genome-wide significant SNPs between the SEM using individual level data and each of the summary statistics methods (Supplementary Data 1 and 2). After Bonferoni correction for the number of SNPs with results, as we would expect, we identified 18 SNPs with significant heterogeneity between the MTAG estimates and the SEM for the offspring effect and 9 SNPs for the maternal effect (3 SNPs showed significant heterogeneity for both the maternal and offspring effect) when there was sample overlap between the GWAS of own and offspring birth weight (Supplementary Data 1). In contrast, were unable to detect significant heterogeneity for any SNPs using mtCOJO (offspring effect $P_{min} = 0.039$, maternal effect $P_{min} = 0.083$), the SEM based on covariance matrices derived from the summary statistics (offspring effect $P_{min} = 0.700$, maternal effect $P_{min} = 0.585$), the linear approximation of the SEM (offspring effect $P_{min} = 0.707$, maternal effect $P_{min} = 0.595$) or Genomic SEM (offspring effect $P_{min} = 0.762$, maternal effect $P_{min} = 0.758$). Similar results were seen when there was no sample overlap between the GWAS of own and offspring birth weight (Supplementary Data 2).

**Table 2 Summary of the results from each of the methods used to derive maternal and offspring specific genetic effects using summary statistics from a GWAS of own birth weight and a GWAS of offspring birth weight.**

| Method | | No sample overlap | | | | | Sample overlap | | | | |
|---|---|---|---|---|---|---|---|---|---|---|---|
| | | Comparison with SEM using individual level data | | Computational time (min)[a] | LD score regression intercept (standard error) | Number of SNPs with $P < 5 \times 10^{-8}$ (Number of loci)[b] [Number of false positive loci][c] | Comparison with SEM using individual level data | | Computational time (min)[a] | LD score regression intercept (standard error) | Number of SNPs with $P < 5 \times 10^{-8}$ (Number of loci)[b] [Number of false positive loci][c] |
| | | Effect estimate | Standard error | | | | Effect estimate | Standard error | | | |
| SEM using summary statistics[d] | Offspring effect | Accurately estimated | Deflated | 4641 | 1.754 (0.012) | 2143 (546) [486] | Accurately estimated | Comparable | 5753 | 1.066 (0.009) | 792 (38) [19] |
| | Maternal effect | Accurately estimated | Deflated | | 1.197 (0.009) | 728 (76) [51] | Accurately estimated | Comparable | | 1.058 (0.008) | 656 (43) [18] |
| Linear approximation of SEM | Offspring effect | Accurately estimated | Comparable | 42 | 1.012 (0.007) | 37 (6) [1] | Accurately estimated | Inflated | 34 | 0.939 (0.008) | 320 (13) [2] |
| | Maternal effect | Accurately estimated | Comparable | | 1.016 (0.007) | 423 (18) [0] | Accurately estimated | Inflated | | 0.941 (0.007) | 496 (16) [0] |
| MTAG | Offspring effect | No correlation with effect estimate from SEM | Deflated | 30 | 0.988 (0.011) | 3595 (80) [5] | No correlation with effect estimate from SEM | Deflated | 30 | 1.001 (0.012) | 5461 (128) [16] |
| | Maternal effect | No correlation with effect estimate from SEM | Deflated | | 1.005 (0.011) | 3880 (89) [11] | No correlation with effect estimate from SEM | Deflated | | 1.005 (0.012) | 5351 (119) [18] |
| mtCOJO | Offspring effect | Consistently underestimated | Deflated | 48 | 1.022 (0.007) | 62 (7) [1] | Consistently underestimated | Deflated | 62 | 1.049 (0.009) | 896 (22) [2] |
| | Maternal effect | Consistently underestimated | Deflated | 36 | 1.031 (0.008) | 386 (17)[0] | Consistently underestimated | Deflated | 63 | 1.043 (0.008) | 524 (18) [0] |
| Genomic SEM[d] | Offspring effect | Accurately estimated | Comparable | 103 | 0.985 (0.007) | 32 (16) [1] | Accurately estimated | Comparable | 224 | 0.971 (0.008) | 394 (13) [2] |
| | Maternal effect | Accurately estimated | Comparable | | 0.978 (0.007) | 372 (16) [0] | Accurately estimated | Comparable | | 0.976 (0.007) | 532 (18) [0] |

Computational time is used as a guide only to compare between methods; this will differ depending on the computing resources available for each analysis.
[a]Analyses were conducted on a Dell Poweredge R840 server that is part of a Rocks 7 open-source Linux cluster based upon CentOS 7.4. Specific details include: CPU: 4 × Intel Xeon Gold 5117 2.0 G, 14 C/28 T, 10.4GT/s, 19.25 M Cache, Turbo, HT (105 W); Disk: 480GB SSD SATA Read Intensive 6 Gbps 512 2.5in; Memory: 24 × 64GB LRDIMM, 2666MT/s, Quad Rank. The number of cores and memory were assigned to each job to optimize performance of the method.
[b]A locus was defined as 500 kb from the sentinel SNP.
[c]A false positive was defined as a locus that is greater than 500 kb from the already known birth weight associated sentinel SNP.
[d]We ran the 22 chromosomes in parallel for the SEM using summary statistics and Genomic SEM, so the computational time was the time to run chromosome 2 as this is the longest chromosome and therefore slowest to complete.

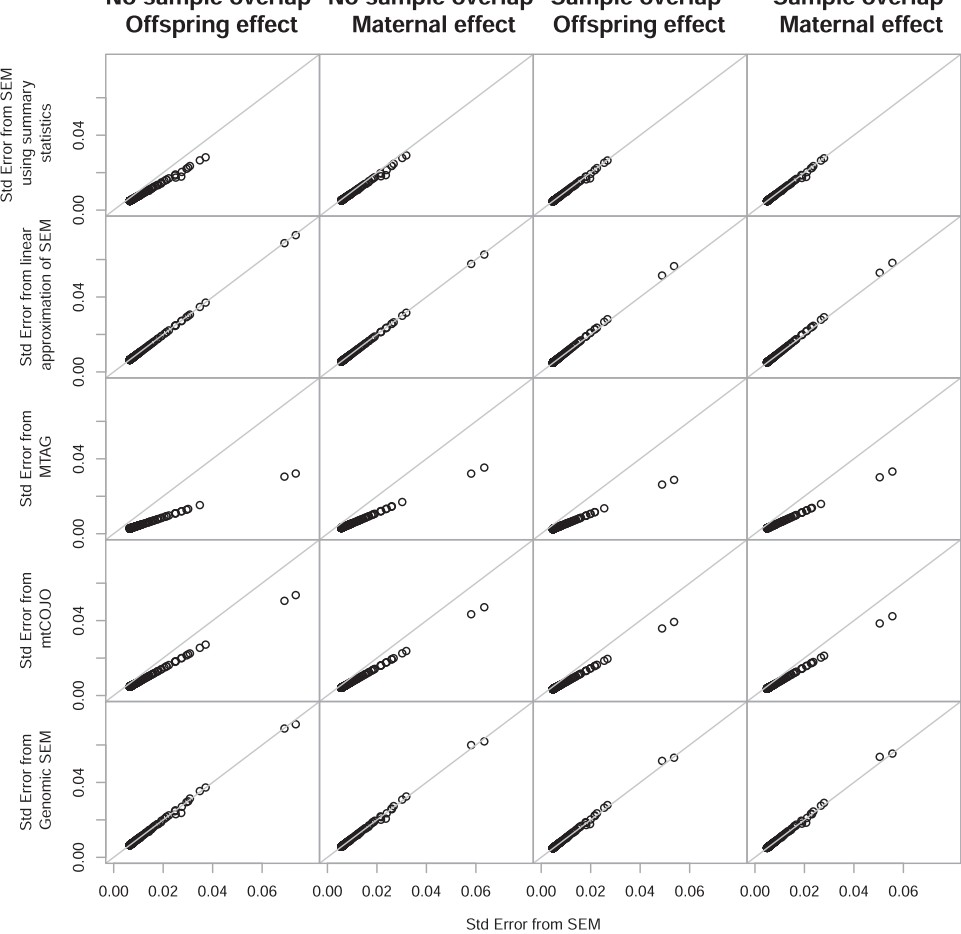

**Fig. 2 Comparison of the standard error from the SEM using individual level data (x-axis) and the various different methods using the summary statistics from the GWAS of own and offspring birth weight (y-axis) for the 300 autosomal genome-wide significant SNPs from Warrington et al.[3].** The columns summarize the results from the analysis including unique individuals in the GWAS of own and offspring birth weight for the offspring and maternal genetic effect, respectively, followed by the results from the analysis where there were overlapping samples in the GWAS of own and offspring birth weight for the offspring and maternal genetic effect.

**Comparison of computational time**. The computational time was not influenced by sample overlap. MTAG, mtCOJO and the linear approximation of the SEM all took approximately between 30 and 60 min (see Table 2 for precise computational time for each method). When running each chromosome in parallel, Genomic SEM took under 4 h to complete. In contrast, the SEM based on covariance matrices derived from the summary statistics took over 60 h to complete, indicating that it is much more computationally intensive than the other methods.

**Evidence of inflation of test statistics across the genome**. Manhattan plots and Q–Q plots for each of the conditional GWAS are presented in Supplementary Figs. 1–20. The LD score regression intercepts and number of genome-wide significant ($P < 5 \times 10^{-8}$) SNPs are presented in Table 2. Inflation in the test statistics was detected for the SEM based on covariance matrices derived from the summary statistics when there was no sample overlap (LD score regression intercepts: offspring = 1.754, maternal = 1.197). This resulted in a larger number of variants/ loci being identified as genome-wide significant (Table 2). This appears to predominantly be driven by the underestimation of the standard error (as seen in Fig. 2), which is most prominent for SNPs with lower minor allele frequency or SNPs where the maternal and offspring genetic effect are going in opposite

directions. Deflation in the test statistics was detected for the linear approximation of the SEM when there was sample overlap (LD score regression intercepts: offspring = 0.939, maternal = 0.941). This is expected as the standard errors are slightly larger because the equations do not account for sample overlap. The LD score regression intercepts for the other methods ranged between 0.971 and 1.066, indicating that there was not much genome-wide inflation in the test statistics.

We defined a birth weight associated locus as being 500 kb from a previously identified birth weight associated sentinel SNP[3]. Although a large number of birth weight associated loci were detected using some of the methods (5–546 loci; Table 2), the majority of them have been previously associated with birth weight (which we assume are true positives). The SEM using summary statistics detected a large number of what are presumably false positives (those loci that have not previously been associated with birth weight in far larger samples of individuals[3]), particularly when there is no sample overlap, which is in line with the inflated LD score intercepts. MTAG also detected a substantial number of false positives, whereas the three other methods only detected up to three false positives.

**Fertility GWAS**. Through the methods comparison, we have shown that Genomic SEM outperforms the other methods in

terms of its ability to accurately estimate conditional effect sizes and standard errors at individual genetic variants and its ability to account for sample overlap appropriately. It is also a highly flexible method, allowing the estimation of other genetic effects, such as paternal-specific effects. To illustrate application of this method and how it can be extended to simultaneously estimate conditional maternal, paternal and offspring genetic effects, we applied it to fertility data from the UK Biobank. Given an off-spring's genotype is correlated ~0.5 with both parental genotypes, and the offspring could influence parental decision to have additional children (for example, due to certain behavioural traits), it is important to adjust for offspring specific genetic effects when investigating the genetic determinants of fertility. We will refer to this offspring specific genetic effect as a sibling-specific effect as we are estimating it using the number of siblings an individual has. Additionally, we will estimate the female-specific genetic effect on fertility using the number of children mothered and the male-specific genetic effect on fertility using the number of children fathered.

Results from the unconditional GWAS analysis conducted in BOLT-LMM of the number of children fathered, the number of children mothered and the number of siblings can be visualized in Supplementary Fig. 21. We conducted two separate analyses in Genomic SEM; firstly we calculated only the female and sibling-specific effects using the GWAS results from number of children mothered and number of siblings (Supplementary Fig. 22), and secondly we utilized all three GWAS to calculate the female, male and sibling-specific effects (Fig. 3). Using LD score regression[18], we estimated the genetic correlations between the unconditional and conditional GWAS. There was a strong genetic correlation between the number of children mothered (unconditional) and the female-specific effect on fertility (analysis one: $r_g = 0.941$, SE = 0.006; analysis two: $r_g = 0.932$, SE = 0.008) and similarly between the number of children fathered and the male-specific effect on fertility (analysis two: $r_g = 0.904$, SE = 0.013). In contrast, the genetic correlation was weaker between the number of siblings and the sibling-specific effect, particularly once male-specific effects were incorporated (analysis one: $r_g = 0.712$, SE = 0.025; analysis two: $r_g = 0.173$, SE = 0.058). We also used LD score regression to estimate the genetic correlation between the conditional GWAS for male and female fertility ($r_g = 0.871$, SE = 0.023), male fertility and sibling effects ($r_g = -0.826$, SE = 0.023) and female fertility and sibling effects ($r_g = -0.812$, SE = 0.019).

Both male and female fertility from the conditional analysis were negatively genetically correlated with years of education (male $r_g = -0.17$, SE = 0.03, $P = 7 \times 10^{-8}$; female $r_g = -0.20$, SE = 0.03, $P = 3 \times 10^{-13}$) and positively genetically correlated with risk-taking behaviours (male $r_g = 0.27$, SE = 0.05, $P = 1 \times 10^{-7}$; female $r_g = 0.17$, SE = 0.04, $P = 6 \times 10^{-5}$; Supplementary Fig. 23), whereas sibling effects were not correlated with years of education or risk-taking behaviours. Additionally, male fertility was negatively genetically correlated with autism spectrum disorder ($r_g = -0.24$, SE = 0.07, $P = 6 \times 10^{-4}$). Sub-jective well-being was also genetically correlated with fertility; positively correlated with male fertility ($r_g = 0.23$, SE = 0.06, $P = 7 \times 10^{-5}$) and negatively correlated with sibling effects ($r_g = -0.23$, SE = 0.06, $P = 3 \times 10^{-4}$).

The results from analysis one estimating the female and sibling-specific effects only on fertility, where we identified four loci ($P < 5 \times 10^{-8}$) associated with the number of children mothered, after conditioning on the number of siblings, and one locus associated with the number of siblings, conditional on the number of children mothered. When we extended the Genomic SEM model to estimate female, male and sibling-specific genetic effects in analysis two, we identified six loci

associated with maternal-specific effects, one locus associated with paternal-specific effects (in the same region as one of the maternal-specific loci) and one locus associated with sibling-specific effects (Fig. 3). After conditioning on male fertility, the locus associated with a sibling-specific effect in the female/sibling only analysis attenuated slightly ($P = 3.6 \times 10^{-5}$), even though it is a different locus to the one identified on chromosome 3 for the male and female-specific effects. The full results for each of these genome-wide significant loci are presented in Supplementary Data 3. Interestingly, a number of the genes nearest to our genome-wide significant loci have previously been associated with age at first sexual intercourse (ESR1, CADM2), number of sexual partners (CADM2), educational attainment (ESR1, TUBB3, MC1R, CADM2, MDFIC) and risk-taking behaviour (MDFIC).

## Discussion

We compared five different statistical methods to estimate maternal and offspring specific genetic effects on an offspring outcome using summary statistics from GWAS and have shown that Genomic SEM outperforms the other methods in terms of accurate estimation of the effect size and standard error, ability to account for sample overlap appropriately, and flexibility to estimate other genetic effects such as a paternal-specific effect. It was more time consuming than several of the other methods; how-ever, running the chromosomes in parallel allowed the GWAS to be completed in under 4 h. Additionally, we detected some deflation in the test statistics that could have been due to the use of the stricter version of genomic control that was implemented in the version of the software used for this analysis; this has been relaxed in more recent releases. Subsequently, we used Genomic SEM to identify the genetic loci associated with male and female fertility, after adjusting for sibling genetic effects, and identified seven loci, one of which was novel.

There are several strengths and limitations of our study. First, not all of the five methods we examined were developed to condition on a correlated genotype (i.e. parental and/or offspring genotype in the present context). In particular, MTAG and mtCOJO are multivariate methods that were specifically devel-oped for other purposes (i.e. to increase power to detect pleio-tropic loci, and to estimate genetic effects conditional on a correlated phenotype, respectively). Previous work has shown that both methods perform excellently when applied to the situations for which they were originally developed[13,16]. However, given the paucity of existing software to estimate conditional effects from summary results data especially genome-wide, we were interested in whether these user friendly software packages could also be used to approximate conditioning on a correlated genotype, and generate accurate parental and offspring specific genetic effects on a phenotype.

Of the comparisons that we made across all the different methods (i.e. comparing effect size estimates and standard errors, computational time, inflation in the test statistics, ability to account for sample overlap and ability to be extended to incor-porate additional genetic effect estimates), Genomic SEM per-formed best on all comparisons except computational time. In contrast, mtCOJO and MTAG did not yield accurate estimates or SEs of conditional maternal and/or offspring genetic effects. Although we based our conclusions on findings from a single large dataset, we believe that our results are likely to hold more generally and are a reflection of Genomic SEM's flexibility in being able to accurately model the relationship between parental and offspring genotypes (i.e. neither MTAG nor mtCOJO spe-cifies this relationship accurately—see below for further discus-sion of this point) and Genomic SEM's ability to take into account sample overlap and cryptic relatedness across the

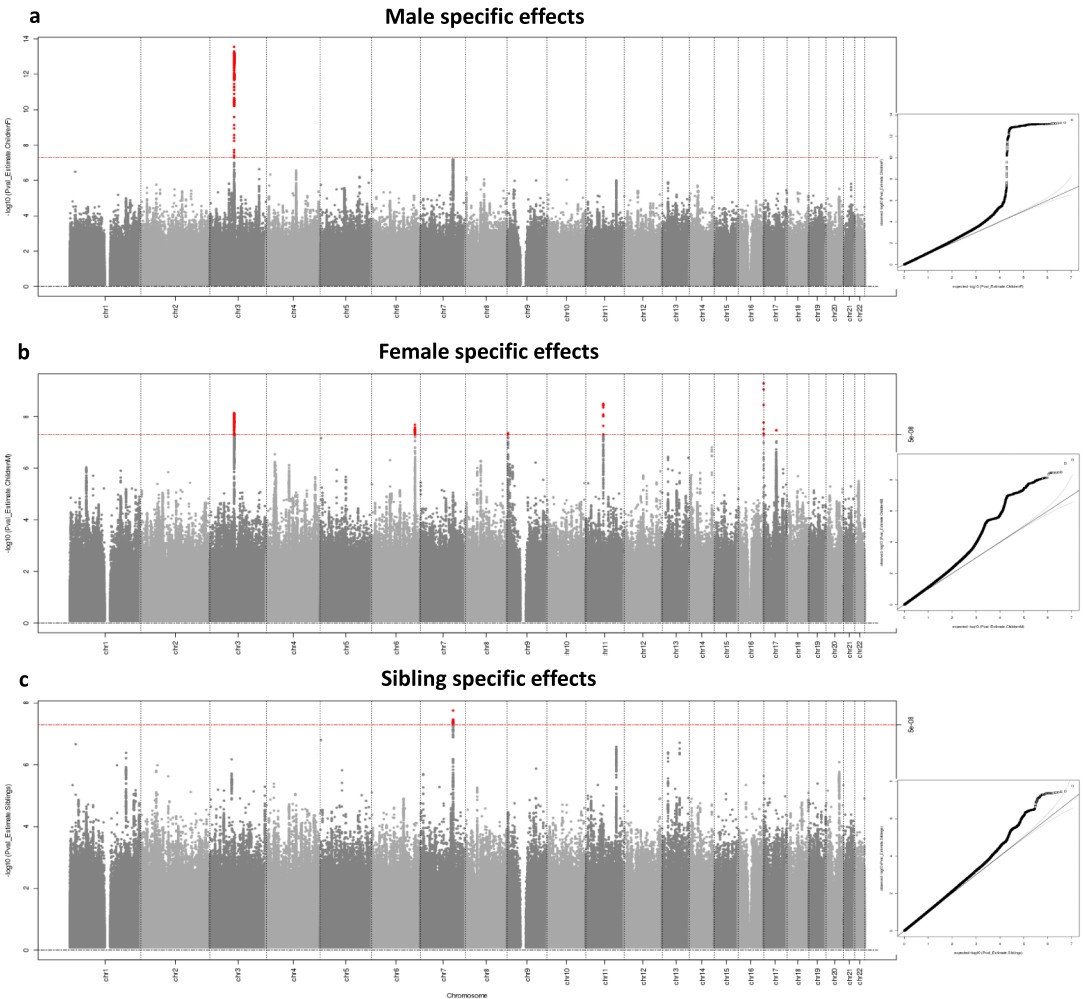

**Fig. 3 Manhattan plot and quantile–quantile (Q–Q) plot for the fertility GWAS estimating male, female and sibling-specific genetic effects using Genomic SEM.** 237,768 women from the UK Biobank contributed to the unconditional GWAS of the number of children mothered, 199,570 men contributed to the GWAS of the number of children fathered and 430,466 individuals contributed to the GWAS of the number of siblings (see Supplementary Fig. 26 for the Manhattan plots of the unconditional GWAS). Point estimates for male, female and sibling effects and their standard errors were estimated using diagonally weighted least squares as implemented in Genomic SEM, and two-sided *P*-values obtained from Z tests on these estimates. The two-sided association *P*-value, on the −log10 scale, obtained from Genomic SEM for each of the SNPs (*y*-axis) was plotted against the genomic position (NCBI Build 37; *x*-axis). Association signals that reached genome-wide significance ($P < 5 \times 10^{-8}$) are shown in red. In the Q–Q plots, the black dots represent observed two-sided *P*-values and the grey line represents expected two-sided *P*-values under the null distribution. The SNP heritability, estimated using LD score regression, was 0.033 (SE = 0.003) for male fertility, 0.042 (SE = 0.003) for female fertility and 0.012 (SE = 0.001) for sibling-specific effects. *P*-values are not adjusted for multiple comparisons.

different GWAS (i.e. the weighted linear model does not estimate sample overlap and utilization of this information is not optimal in ordinary SEM).

There were several limitations with using summary statistics in the SEM, which resulted in inflation in the test statistics. First, we estimated the sample overlap using LD score regression[18], which was overestimated in both of our analyses with and without sample overlap. This overestimation has been described previously when there is population stratification[19], and the authors suggest a modified formula to calculate the overlap. We did not use this modified formula in the current analysis as we and others have previously used LD score regression to estimate sample overlap and we wanted to get an idea of how this would perform[3,20]. It could also be due to cryptic relatedness between the GWAS; for example, there might be close relatives in both GWAS of own and offspring birth weight that are adding to this overestimation. Second, estimates of the maternal and offspring specific genetic effects can vary dramatically if the phenotypic

correlation between the maternal and offspring phenotype is misspecified (results not shown). Given we had access to the phenotypic data that was used for the GWAS analysis, we were able to obtain a good estimate of the phenotypic correlation; however, this might be more difficult to estimate accurately if using publically available GWAS results. Third, this method assumes that the effect size estimates from the unadjusted GWAS are estimated accurately and does not account for the standard errors. Therefore, for low frequency genetic variants that have large standard errors, we saw the method performed poorly. We also found the method performed poorly for a subset of genetic variants where the maternal-specific genetic effect on the offspring outcome went in the opposite direction to the offspring specific genetic effect, particularly when there was no sample overlap between the unadjusted GWAS. However, as we showed in the initial paper describing the SEM[9], including some raw data in addition to the covariance matrices estimated from the summary statistics improves estimation of the maternal and offspring

specific genetic effects. It is likely that the SNPs that reached genome-wide significance using this method, but are unknown birth weight associated loci, are false positives for three main reasons; (1) as seen in the Manhattan plots presented in the Supplementary Material, the majority of these SNPs are singletons and not part of LD blocks, (2) none of the other methods included in the comparison identified these loci and (3) the most recent GWAS of birth weight[3], which included the data in this study in addition to data from many birth cohorts and also partitioned the genetic effect into maternal and offspring components, did not identify these loci.

The linear approximation of the SEM assumes no sample overlap, so performed well when there was no overlap; however, the standard errors were overestimated when there was sample overlap, deflating the test statistics. The formula for estimating the standard error of the maternal and offspring specific effects could be extended to account for any sample overlap, but it would need to rely on LD score regression to estimate the sample overlap which has issues as described previously. Otherwise, this method performed well in terms of accurately estimating the maternal and offspring specific effects and was one of the fastest methods to perform the conditional analysis.

Given that MTAG is not designed to estimate maternal and offspring specific effects at individual loci, it was not surprising that it performed poorly in terms of accurately partitioning the genetic effect into maternal and offspring components. This is because the MTAG model estimates combined pleiotropic genetic effects on both phenotypes (i.e. in the context of this manuscript, a pleiotropic effect on own birth weight and offspring birth weight c.f. Supplementary Fig. 24). Intuitively, the MTAG model borrows power from a correlated phenotype to increase overall power to detect association. Previous work by other groups suggests that MTAG is likely to be a powerful approach if the goal of the investigator is locus discovery- particularly in situations where the magnitude of the genetic correlation between variables is high, and where the pattern of genetic effects at the individual SNP level is concordant with the genetic correlation between the phenotypes across the genome more broadly[13]. In contrast, our results suggest that if the focus is on locus characterization/ accurately partitioning effects into maternal and offspring components (e.g. for downstream MR analyses where it is important to block potentially pleiotropic paths through related individuals[10]) then one of the SEM based procedures discussed in this manuscript will be more appropriate.

Finally, mtCOJO was originally developed to condition the outcome on one or more covariate phenotypes[16]; whereas the other methods we compare are equivalent to conditioning the outcome on the genotype. For example, to estimate the offspring specific genetic effect, mtCOJO is conditioning the genetic effect on offspring birth weight rather than maternal genotype. This means that the effects estimated by mtCOJO were slightly different to those obtained using the SEM based approaches. We therefore recommend that when the goal is to accurately estimate maternal and offspring genetic effects (e.g. for downstream Mendelian randomization analyses) that other methods be used.

We performed the first GWAS partitioning the genetic effect into male and female fertility specific genetic effects and a sibling-specific effect. The heritability of number of children ever born has been estimated to be between 0.24 and 0.43[21], and the variance explained by common genetic variants (SNP based heritability) has been estimated to be ~10%[22]. Although there have been two GWAS previously conducted on number of children born[23,24], no study to date has estimated the conditional male, female and sibling genetic effects at individual genetic loci. Both previous studies observed significant genetic correlations for the number of children between men and women (Barban and

colleagues[24]: $r_g = 0.97$, SE = 0.095; Mathieson and colleagues[23]: $r_g = 0.74$, 95% CI = 0.66–0.82), which our findings from the conditional analysis are consistent with ($r_g = 0.871$, SE = 0.023). We also identified a strong negative genetic correlation between both male/female fertility and sibling effects; this is likely to be due to a technical artefact of the analyses as described by Wu and colleagues[25]. We replicate the negative genetic correlation between years of education and fertility described in Barban et al.[24]. Furthermore, we found a positive genetic correlation between risk-taking behaviour and both male and female fertility, showing that having more increasing alleles for the number of children is associated with a higher genetic risk for partaking in risk-taking behaviours. Due to our ability to partition the genetic effect, we were also able to identify a negative genetic correlation between male fertility and autism, indicating that fathers at genetically increased risk of autism are more likely to have fewer children. This relationship between fertility and autism, particularly in males, has previously been shown in a large population based study in Sweden using patients with various psychiatric disorders and their unaffected siblings[26]. Interestingly, we identified a positive genetic correlation between male fertility and subjective well-being and a negative genetic correlation with the sibling effect. We identified six of the previous 28 statistically independent loci for fertility, and one of the 16 loci for childlessness[23]. In addition, we identified a novel locus on chromosome 9 that is associated with female fertility. This locus is near *RFX3*, which harbours genetic variants that have previously been associated with smoking initiation. Our genetic correlation analysis shows a positive genetic correlation between both male and female fertility and smoking initiation; however, it does not meet our multiple testing threshold.

In conclusion, when estimating maternal and offspring (and paternal) specific genetic effects on an offspring outcome using GWAS summary statistics, we recommend using Genomic SEM.

## Methods

**Participants.** The UK Biobank has ethical approval from the North West Multi-Centre Research Ethics Committee (MREC), which covers the UK, and all participants provided written informed consent. UK Biobank phenotype data was available on 502,543 individuals, of which 280,142 reported their own birth weight at either the baseline or first two follow-up visits. There were 7701 individuals who were part of a multiple birth and were excluded from the analyses. There were 10,670 individuals who reported their birth weight at more than one visit, with 83 individuals reporting the two values to be different by more than 1 kg; these individuals were excluded from the analyses. For those individuals who reported different values between baseline and follow-up (<1 kg) we took the measure from the first reported visit for the analyses. Finally, we excluded individuals who reported their birth weight to be <2.5 kg or >4.5 kg, as these are implausible for live term births before 1970. In total, 234,154 individuals had data on their birth weight matching our inclusion criteria.

Women in the UK Biobank were also asked to report the birth weight of their first child to the nearest pound and were converted to kilograms for analyses (N = 216,782). We excluded individuals with multiple measures that differed by >1 kg (N = 29) or if their birth weight was <2.2 kg (5 pounds) or >4.6 kg (10 pounds), leaving 210,423 individuals with birth weight of their first child matching our inclusion criteria.

We analysed genetic data from the April 2018 release of imputed data from the UK Biobank, a resource that is described extensively elsewhere[27]. In addition to the quality control metrics performed centrally by the UK Biobank, we defined a subset of white European ancestry, unrelated individuals. First, we generated ancestry informative principal components (PCs) in the 1000 genomes samples. The UK Biobank samples were then projected into this PC space using the single nucleotide polymorphism (SNP) loadings obtained from the PC analysis using the 1000 genomes samples. The UK Biobank participants' ancestry was classified using K-means clustering centered on the three main 1000 genomes populations (European, African, South Asian). Those clustering with the European cluster were classified as having European ancestry. The UK Biobank participants were asked to report their ethnic background. Only those reporting as either "British", "Irish", "White" or "Any other white background" were included in the clustering analysis. Second, to identify a subset of unrelated individuals in the UK Biobank, we generated a genetic relationship matrix in the GCTA software package[28] (version 1.90.2) and excluded one of every pair of related individuals with a genetic

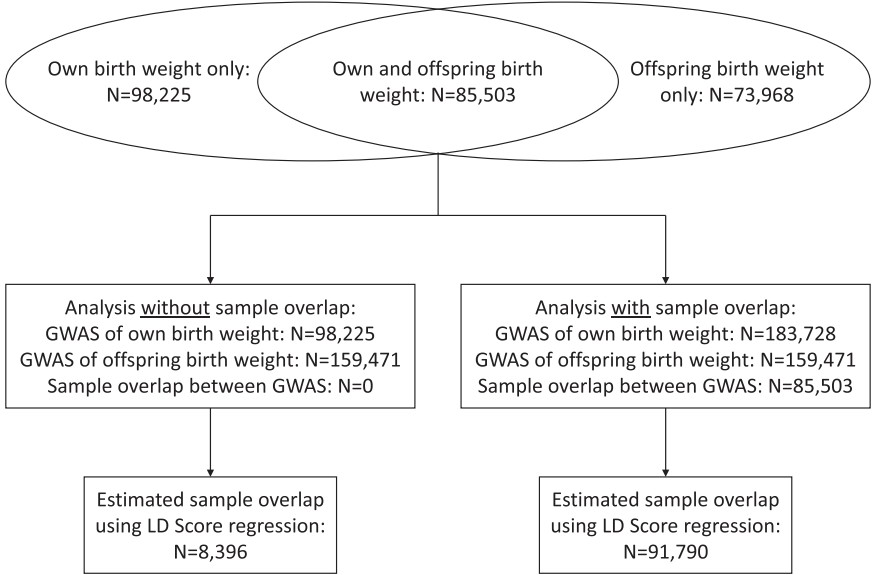

**Fig. 4 Schematic of the study design for comparing methods using self-reported birth weight data from the UK Biobank.** We conducted two sets of analysis, one with and one without sample overlap between the genome-wide association studies (GWAS), to investigate the effect of sample overlap in each of the methods.

relationship greater than 9.375%. A subset of 257,696 individuals with genotype data, a valid birth weight for themselves or their first child, were unrelated and were genetically of white European ancestry remained for analysis. Of these, 72,274 were men so only reported their own birth weight, 25,951 women reported only their own birth weight, 73,968 reported only the birth weight of their first child and 85,503 reported both. We adjusted both the individuals' own birth weight and the birth weight of their first child for the principal components provided by the UK Biobank, assessment center and genotyping array, and sex for own birth weight, and then created z-scores.

Because we were interested in how each of the statistical methods handled sample overlap, we created two sets of data (Fig. 4). The first contained all of the data available, including the 85,503 that contributed to both the GWAS of own birth weight ($N = 183,728$) and the GWAS of offspring birth weight ($N = 159,471$). The second contained all of the data for the GWAS of offspring birth weight ($N = 159,471$) but excluded those individuals from the GWAS of own birth weight that were included in the GWAS of offspring birth weight ($N = 183,728 - 85,503 = 98,225$).

**GWAS analysis**. GWAS of own and offspring birth weight was conducted using a linear mixed model implemented in BOLT-LMM v2.3.2[29] to account for population structure and subtle relatedness. Only autosomal genetic variants which were common (minor allele frequency >0.1%), had Hardy-Weinberg equilibrium P-value > $1 \times 10^{-6}$ and missingness <0.1 were included in the genetic relationship matrix (GRM). We excluded genetic variants with an INFO score < 0.4 and minor allele frequency <0.1% from the analysis in BOLT-LMM (BOLT-LMM uses the full sample to exclude SNPs based on these thresholds, so some SNPs may have minor allele frequency <0.1% in our subset of the UK Biobank data with clean birth weight data). We then used the summary statistics from the GWAS of own and offspring birth weight in the following analyses to estimate the conditional maternal and offspring genetic effects at each genetic variant.

**SEM analysis using summary statistics**. The SEM to estimate the adjusted maternal and offspring genetic effects has been described in detail previously[9] (Supplementary Fig. 25). Briefly, to estimate the parameters for the adjusted offspring and maternal genetic effects on birth weight, we use three observed variables available in the UK Biobank; the participant's genotype, their own self-reported birth weight, and in the case of the UK Biobank women, the birth weight of their first child. Additionally, the model comprises two latent (unobserved) variables, one for the genotype of the UK Biobank participant's mother and one for the genotype of the participant's offspring. From biometrical genetics theory, these latent genetic variables are correlated 0.5 with the participant's own genotype, so we fix the path coefficients between the latent and observed genotypes to be 0.5. We have previously described how the SEM can be fit with either the individual level data or observed covariance matrices derived from the individual level data[9]. To derive the observed covariance matrices from GWAS summary statistics, we need the allele frequency of the genetic variant, beta coefficient from the regression model of the genetic variant on own or offspring phenotype, variance of the phenotype (which will be one if the phenotype was standardized prior to the regression analysis) and the sample size. We assume that the allele frequency, beta

coefficient and variance is consistent across the following three groups, but the sample size will differ: individuals with their own phenotype only, individuals with their offspring's phenotype only and individuals with both. We therefore need to estimate the sample overlap from the summary statistics in order to include the sample size for each of the covariance matrices. To do this, we performed bivariate linkage disequilibrium (LD) score regression (version 1.0.0) analysis using the summary statistics from the GWAS of own birth weight and the GWAS of offspring birth weight and used the regression intercept to estimate the number of individuals in both analyses. We used a phenotypic correlation between own and offspring birth weight of 0.24 in the calculation, which was estimated using the cleaned phenotype data that was included in the GWAS analysis. We have previously shown that the SEM has difficulty optimizing with low frequency variants, so we excluded SNPs with a minor allele frequency less than 0.5%. For each genetic variant we then calculated the observed covariance matrices from the summary statistics and fit the SEM with the relevant estimated sample sizes. We calculated a Wald P-value for the maternal and offspring genetic effects using the effect size estimates and their standard errors. We conducted the analysis of each chromosome in parallel to reduce the computational time. Analyses were conducted in R (version 3.4.3) using the OpenMx package (version 2.6.9).

**Analysis using a linear approximation of the SEM**. We have previously derived a weighted linear model that is a good approximation of the SEM but substantially less computationally intensive[3]. This model uses a linear transformation of the effect sizes from the GWAS of own birth weight and the GWAS of offspring birth weight based on the principles of ordinary least squares linear regression. The offspring effect at each genetic variant is estimated as:

$$\hat{\beta}_{o_{adj}} = -\frac{2}{3}\hat{\beta}_{m_{unadj}} + \frac{4}{3}\hat{\beta}_{o_{unadj}} \qquad (1)$$

And the corresponding standard error is:

$$\mathrm{SE}\left(\hat{\beta}_{o_{adj}}\right) = \sqrt{\frac{4}{9}\mathrm{var}\left(\hat{\beta}_{m_{unadj}}\right) + \frac{16}{9}\mathrm{var}\left(\hat{\beta}_{o_{unadj}}\right)} \qquad (2)$$

Where $\hat{\beta}_{(o\_adj)}$ is the offspring effect adjusted for the effect of maternal genotype, $\hat{\beta}_{(m\_unadj)}$ is the unadjusted maternal effect from the GWAS of offspring birth weight and $\hat{\beta}_{(o\_unadj)}$ is the unadjusted offspring effect from the GWAS of own birth weight. Likewise, the maternal effect is estimated as:

$$\hat{\beta}_{m_{adj}} = \frac{4}{3}\hat{\beta}_{m_{unadj}} - \frac{2}{3}\hat{\beta}_{o_{unadj}} \qquad (3)$$

And standard error is:

$$\mathrm{SE}\left(\hat{\beta}_{m_{adj}}\right) = \sqrt{\frac{16}{9}\mathrm{var}\left(\hat{\beta}_{m_{unadj}}\right) + \frac{4}{9}\mathrm{var}\left(\hat{\beta}_{o_{unadj}}\right)} \qquad (4)$$

Where $\hat{\beta}_{(m\_adj)}$ is the maternal effect adjusted for the effect of offspring genotype. The full derivation can be found in Warrington et al. (2019)[3]. Similar to the SEM using summary statistics, we calculated a Wald P-value for the maternal and offspring genetic effects using the effect size estimates and their standard errors. This method assumes that the two unadjusted GWAS are independent, and

do not contain any sample overlap. We performed the linear transformation in R (version 3.4.3) and each of the chromosomes were run in parallel to reduce the computational time.

**Analysis using MTAG.** MTAG enables joint analysis of multiple traits using summary statistics from GWAS, while accounting for the possibility of overlapping samples, and produces trait-specific effect estimates for each genetic variant[13]. It is based on the idea that when GWAS estimates from different traits are correlated, the SNP effect estimates can be improved by incorporating information from the other correlated traits. This method was not developed to estimate SNP effects conditional on parental genotypes; however, we were interested in investigating how well it approximated the maternal and offspring genetic effects on birth weight (Supplementary Fig. 24). We used python version 2.7.12 and set the lower sample size bound to zero (--n_min 0.0) when running MTAG.

**Analysis using mtCOJO.** mtCOJO[16] performs an approximate multi-trait-based conditional GWAS using summary statistics from a GWAS of two or more traits. For our birth weight analysis, the method approximates the following two models (Supplementary Fig. 26):

$$BW = \beta_{off}BW_{off} + \beta_{SNP\_own,i}SNP_i + \varepsilon \tag{5}$$

$$BW_{off} = \beta_{own}BW + \beta_{SNP\_off,i}SNP_i + \varepsilon_{off} \tag{6}$$

where $BW$ is the individual's own birth weight, $BW_{off}$ is offspring birth weight, $SNP_i$ is the ith SNP, $\beta_{off}$ is the effect of offspring birth weight on the an individual's own birth weight, $\beta_{SNP\_own,I}$ is the effect of $SNP_i$ on the individual's own birth weight, $\beta_{own}$ is the effect of individual's own birth weight on the their offspring's birth weight, $\beta_{SNP\_off,i}$ is the effect of $SNP_i$ on offspring birth weight, $\varepsilon$ and $\varepsilon_{off}$ are the residuals. Although this method conditions on the phenotype of the other individual in the pair (i.e. conditioning on the offspring's phenotype when analysing own birth weight, rather than their genotype), we wanted to investigate how well this approach would approximate conditioning on the genotype. In other words, we were interested in investigating how well $\hat{\beta}\_(SNP\_own,i)$ approximates $\hat{\beta}\_o$ from the SEM for SNP $i$, and $\hat{\beta}\_(SNP\_off,i)$ approximates $\hat{\beta}\_m$.

To conduct analysis in mtCOJO, we needed a reference sample with individual level genotypes for LD estimation. Therefore, we randomly sampled 50,000 individuals from the UK Biobank and extracted their imputed genetic data, using Plink2 (released 18 March 2019), for each of the unique genetic variants that were included in the cleaned GWAS summary statistics. mtCOJO could not handle genetic variants that had the same rs number but different alleles (i.e. multi-allelic markers), so we removed all duplicate rs numbers from the reference dataset. Using this reference dataset, we conducted the mtCOJO analysis with the default parameters in GCTA (version 1.92.0beta3), using the summary statistics from the GWAS of own birth weight as the outcome and conditioning on offspring birth weight and then using the summary statistics from the GWAS of offspring birth weight as the outcome and conditioning on own birth weight.

**Analysis using Genomic SEM.** Genomic SEM[17] is a highly flexible, two stage multivariate statistical method for analysing the joint genetic architecture of traits using GWAS summary results statistics. In stage one, a $K$ order genetic covariance matrix is estimated from the genome-wide summary results data of $K$ GWAS using LD score regression[17]. Estimates of the standard errors for each of the variance-covariance terms, which account for sample overlap between the GWAS are also obtained. This stage contrasts with ordinary structural equation modelling, which uses a covariance matrix obtained from individual level data (e.g. a covariance matrix derived from individual level genotype, own birth weight and offspring birth weight). In stage two, a user specified model is then fit to the genetic covariance matrix in an attempt to explain the underlying pattern of genetic correlations across the traits in terms of a series of latent genetic variables. The model can be augmented through the addition of observed SNP variables, providing the opportunity to perform multivariate tests of association between individual SNPs and phenotypes, estimate the conditional effect of SNPs, and in some cases increase statistical power to detect association. In this manuscript, we create a path model based on standard biometrical genetics theory to model the genetic relationship between own and offspring birth weight, and use Genomic SEM to estimate conditional maternal (paternal) and offspring specific genetic effects. The specific model that we fit to the birth weight data is depicted in Supplementary Fig. 27.

We conducted GWAS analysis using the userGWAS function in Genomic SEM v0.0.2 (installed 9 Jan 2020), which creates genetic covariance matrices for individual SNPs and estimates SNP effects for a user specified multivariate GWAS. Following the workflow described on the github Wiki, we ran multivariate LD score regression to estimate the genetic covariance matrix and corresponding sampling covariance matrix, which accounts for any potential sample overlap between the GWAS summary statistics. After preparing the summary statistics for analysis, we used the estimated matrices to run the GWAS using 50 cores on a computing cluster.

**Comparison of methods.** We fit the same SEM as described above, but using the individual level data rather than observed covariance matrices, for the 300 autosomal genetic variants that reached $P < 5 \times 10^{-8}$ in the latest GWAS of birth weight[3]; we excluded rs2428362 from the comparison as it is tri-allelic. For the SEM using the data with no sample overlap (i.e. where the individuals from the UK Biobank had either their own birth weight measure or their offspring's, but not both), we did not estimate the correlation between the birth weight measures as we had no data to estimate the parameter. We visualized the difference between the effect size estimates and standard errors from this SEM and those estimated using the GWAS summary statistics and methods described above. We conducted a heterogeneity test to assess the difference in the beta coefficients using the rmeta package (version 3.0) in R (version 3.5.2).

We were also interested in how the methods compared in terms of computational time, inflation of the test statistics and number of genome-wide significant SNPs identified. We saved the computational time for each of the methods (we used the run time from chromosome two for the time of the SEM using summary statistics as each chromosome ran in parallel so this was the longest chromosome to run) for comparison purposes. We note that the computational time will differ between computing resources and we present them here to compare the methods relative to each other. We conducted an LD score regression (version 1.0.0) analysis to estimate the inflation in test statistics for each method.

**Application to real data: fertility GWAS.** In the UK Biobank, 272,579 women and 225,349 men reported how many children they had given birth to (live births only) or fathered, respectively. Additionally, each of the participants reported how many full brothers ($N = 493,181$) and sisters ($N = 493,257$) they had at each follow-up. There were 28,609 women who reported how many children they mothered at more than one visit, 263 (0.9%) of whom changed their response over time so were excluded. Similarly, there were 26,171 men who reported how many children they had fathered at more than one visit, 1103 (4%) of whom changed their response over time so were excluded. In terms of siblings, 54,480 participants reported how many full brothers they had and 54,489 how many full sisters they had at more than one visit, with 2430 (4%) and 1802 (3%) excluded, respectively, because their response changed over time. We added the number of brothers and sisters to get the total number of siblings, with 489,701 participants reporting how many siblings they had available for analysis. Participants reporting greater than 10 siblings ($N = 1,720$, 0.4%) or children (mothered $N = 18$, 0.007%; fathered $N = 43$, 0.02%) were recoded to have 10 in case these were data errors and to avoid a distribution with a large tail. We excluded individuals who were not part of our white European ancestry cluster, leaving 237,768 women reporting how many children they mothered, 199,570 men reporting how many children they had fathered and 430,466 individuals reporting how many siblings they have available for GWAS analysis.

GWAS of the number of siblings and number of children for the women and men were conducted using a linear mixed model implemented in BOLT-LMM v2.3.2[29]. We used the same GRM as was used in the birth weight analyses, and excluded genetic variants with an INFO score <0.4 and minor allele frequency <0.1% from the analysis. We adjusted for the 40 principal components provided by the UK Biobank, genotyping array, age that the number of siblings or children was reported, assessment centre that the participant attended, and for the siblings analysis we also adjusted for sex of the participant. Subsequently, we used the summary statistics to conduct two analyses in Genomic SEM;

(1) Similar to the birth weight analysis, we used the GWAS summary statistics of the number of siblings and number of children mothered to generate female and sibling-specific genetic effects on fertility (Supplementary Fig. 28A).

(2) We used the GWAS summary statistics for all three traits to generate female, male and sibling-specific effects for fertility to illustrate how the structural equation model can be extended when data from fathers is also available (Supplementary Fig. 28B).

To investigate how Genomic SEM performed on a genome-wide scale, we estimated genetic correlations between the unconditional GWAS conducted in BOLT-LMM and the conditional GWAS conducted in Genomic SEM using LD score regression[18]. Subsequently, we used LD Hub[30] (ldsc.broadinstitute.org) to estimate genetic correlations between the conditional estimates from Genomic SEM and a range of developmental, reproductive, behavioural, neuropsychiatric and anthropometric phenotypes that were investigated in Barban et al.[24]. We also investigated the genetic correlation between the conditional estimates and risk-taking behaviour as one of the genome-wide significant loci was previously associated with this trait. Due to the different linkage disequilibrium structure across ancestry groups, we only used summary statistics from LD Hub that were of European origin. There were several traits that had summary statistics from multiple GWAS available in LD Hub, so we used the latest GWAS to estimate the genetic correlations with fertility.

**Reporting summary.** Further information on research design is available in the Nature Research Reporting Summary linked to this article.

## Data availability

Human genotype and phenotype data on which the results of this study were based were accessed from the UK Biobank (http://www.ukbiobank.ac.uk/) with accession ID 53641.

The genotype and phenotype data are available upon application from the UK Biobank (http://www.ukbiobank.ac.uk/). GWAS summary statistics from the fertility GWAS are available at the Evans Group website (https://evansgroup.di.uq.edu.au/GWAS_RESULTS/FERTILITY/). Genomic positions are based on NCBI Build 37.

## Code availability

All analyses conducted in this manuscript were performed with publicly available software or published code.

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

## Acknowledgements

This research was carried out at the Translational Research Institute, Woolloongabba, QLD 4102, Australia. The Translational Research Institute is supported by a grant from the Australian Government. This study has been conducted using the UK Biobank Resource under Application Number 53641. D.M.E. is supported by an Australian National Health and Medical Research Council Senior Research Fellowship (1137714) and this work was supported by a Australian National Health and Medical Research Council Project Grant (1157714) and Ideas Grant (1183074). M.G.N. is supported by the Jacobs foundation, ZonMW grants 849200011 and 531003014 from The Netherlands Organisation for Health Research and Development, and a VENI grant awarded by NWO (VI.Veni.191 G.030).

## Author contributions

N.M.W., M.G.N. and D.M.E. conceived the study. N.M.W. and L.D.H. performed data analysis. N.M.W. and D.M.E. wrote the manuscript. All authors revised and reviewed the paper.

## Competing interests

The authors declare no competing interests.
