## [Peer Review File · Nature Communications]

Estimating direct and indirect genetic effects on offspring phenotypes using genome-wide summary results dataReviewers' Comments:

Reviewer #1:

Remarks to the Author:

This is an interesting paper assessing different methods of doing structural equation modelling (SEM) using genetic summary statistics. Specifically, the authors test 4 different methods using genetic summary data against a "gold standard" SEM using individual data to separate out maternal and own genetic effects on birthweight. They then provide an example of using the best performing summary method to partition genetic associations with fertility.

Introduction

It would be very helpful to provide directed acyclic graphs (DAG) showing the causal structure of the questions addressed by each of these 4 summary data methods in contrast to the DAG for the individual method.

Please give the intuitive explanation and underlying principle motivating each of the summary data methods used to approximate individual SEM. Specifically, do the different summary data methods represent different ways of using the same data which would all be expected to converge on the same estimates? It looks like MTAG and mtCOJO are not quite estimating the same quantities as the two SEM based methods (linear approximation to SEM and Genomic SEM). Is that correct? If so is it worth comparing MTAG and mtCOJO with the SEM based methods?

Methods

A summary table giving the different methods, their assumptions, the data used, parameters used and the tests conducted for each method would be very helpful.

Why was the simulation under the null only carried out for mtCOJO?

Results

Again a summary table would be helpful.

Discussion

Please focus in the discussion on describing the strengths and limitations of this study, i.e., comparing 4 methods for genetic summary data to obtain SEM estimates against one individual SEM method addressing the same question. For example, are all the 4 summary methods comparable in concept? Are one or two comparisons sufficient to say one method is better than another? At the limit are all methods equivalent to the individual level SEM? Why does Genomic SEM do better than the other methods? Is it a general property of Genomic SEM or was it a chance finding in this particular example? Is Genomic SEM better than the linear approximation to SEM because it does not use the fixed constants given in the equations for the linear approximation to SEM?

Reviewer #2:

Remarks to the Author:

Warrington et al present a comparison of published methods for separating direct and indirect genetic effects on offspring phenotypes using summary GWAS results. They find that Genomic SEM performs most consistently across their tests and recommend it as the preferred method for estimating independent parental and offspring association estimates. Separation of independent parental and offspring genetic associations is an important problem in GWAS. My comments on the manuscript are listed below.

In the methods section the authors refer to the standard errors of the SEM using covariance matrices

as under estimated when there is no sample overlap, however in the subsequent sentence they state that this could be due to the presence of sample overlap not accounted for in the SEM using individual level data. These two sentences appear conflicting – it could be that the SEM using individual level data is over estimating the standard errors – can the authors clarify these sentences and justify why they describe the standard errors as under estimated?

They also describe the standard errors from mtCOJO as overestimated and underestimated when discussing the results of simulations – while the simulations suggest the standard errors are in fact overestimated under the null, could the authors justify why they describe the standard errors as underestimated for the 300 loci previously associated with birth weight rather than for example the possibility that conditioning on phenotype rather than genotype increases power at these loci.

When examining the summary statistics across the genome the SEM using summary data finds a large number of genome wide significant loci compared to the other methods, many of which have not previously been associated with birth weight which are described as false positives. Given the inflation in the test statistics many of these are likely to be false positives, however the authors state that standard errors are “underestimated” for SNPs where maternal and offspring genetic effects are in opposite directions. These loci are less likely to be discovered in unadjusted GWAS of birth weight – it would help the reader to discuss how likely it is that discovery of SNPs with opposite maternal and fetal directions which may be masked in previous GWAS contribute to these genome-wide significant SNPs given the difference in sample size between previous GWAS of birth weight and the current analysis.

In the discussion the authors suggest that MTAG is the most powerful method for detecting novel loci as its effect estimates are the sum of maternal and offspring effects at that locus. The increased power is only true when effects are in the same direction – there will be cases where maternal and offspring effects are in opposite directions and the sum of the effects is zero. In these cases MTAG would have reduced power to detect associations and the other methods described would have more power than MTAG to detect these loci.

Reviewer #3:

Remarks to the Author:

The Authors start their journey from the problem of estimating the effect of an individual's own genotype G on their own phenotype Y as separate from the effect of the maternal genotype G_g on Y . Then they review a SEM model that tackles the problem by either using individual-level data or observed covariance matrices derived from such data. Part of the methodology is devoted to circumventing the bias due to overlap between subsamples of individuals with different missingness patterns. All this methodology has been previously introduced by the Authors elsewhere, but the present paper provides additional practical/computational guidance in its use. The topic is very important, and relevant the many situations where one needs to dissect concurrent sources of genetic effect.

In a second part of their paper, the Authors describe a simulation study to evaluate a number of recent established alternative approaches to the above SEM approach, that have the computational advantage of working at a summary statistics level and may or may not deal with overlap (including a weighted linear approximation of the SEM). They comparatively evaluate the method against the results from the application of their SEM, based on individual data. The Genomic SEM approach turns out to be the winner. The Authors apply this method in a study in which, for the first time, they estimates the conditional male, female and sibling genetic effects at individual genetic loci. They discover new loci associated with maternal effects.

The value of this paper does not seem to rest on original methodology. The paper is of great interest

to scientists involved in studies where genetic effect decomposes into parent-specific components, as well as to the wider community of researchers in genomic epidemiology, and moreover to researchers intending to adopt the described methods as a first stage of a rigorous Mendelian randomization study.

THEREFORE I AM IN FAVOUR OF THE SUBMITTED PAPER BEING ACCEPTED FOR PUBLICATION ON THIS JOURNAL.

In the case of a paper revision, my comments are:

- 1) slightly more space should be devoted to explaining the principles of the Genomic SEM approach (lines 252 --)
- 2) at lines 488--, before launching into a description of the computational strategy/procedure, some readers might welcome a description of the problem in scientific terms (defining the three effects components), and a brief discussion helping their intuition of how the problem is "translated" into a SEM model.
- 3) the Discussion should clearly distinguish which considerations are a direct output of the submitted paper, and which are reported from previous studies.

We would like to sincerely thank each of the reviewers for their review of our manuscript. Below is a response to each of the concerns they raised.

Reviewer #1:

- 1. Introduction: It would be very helpful to provide directed acyclic graphs (DAG) showing the causal structure of the questions addressed by each of these 4 summary data methods in contrast to the DAG for the individual method.**

Response: We have included path diagrams and explanations of each of the methods in the supplementary material (Supplementary Figures 1-4) and refer to them in the relevant methods sections.

- 2. Introduction: Please give the intuitive explanation and underlying principle motivating each of the summary data methods used to approximate individual SEM. Specifically, do the different summary data methods represent different ways of using the same data which would all be expected to converge on the same estimates? It looks like MTAG and mtCOJO are not quite estimating the same quantities as the two SEM based methods (linear approximation to SEM and Genomic SEM). Is that correct? If so is it worth comparing MTAG and mtCOJO with the SEM based methods?**

Response: The reviewer is correct that MTAG and mtCOJO are not quite estimating the same quantities as the SEM based methods (see Supplementary Figures 1-4); however, we were interested in how well they approximated the conditional genetic effect estimates obtained under the SEM using individual level data and their standard errors as (i) there is a dearth of software in the genetics community that will generate conditional estimates of maternal and offspring genetic effects from summary results data- especially on a genome-wide scale, (ii) both MTAG and mtCOJO are implemented in user friendly software packages and (iii) they are computationally much more efficient than the SEM based models. We have provided some additional intuition at the beginning of the methods section where we describe which methods we have chosen for comparison (new text is underlined):

“In addition to our published structural equation model (SEM)⁹ and linear approximation of the SEM³, we identified three published methods including multi-trait analysis of GWAS (MTAG)¹³, multi-trait-based conditional and joint analysis using GWAS summary data (mtCOJO)¹⁶ and Genomic SEM¹⁷. MTAG is a multivariate method which uses genome-wide GWAS summary results from multiple correlated phenotypes to increase power to detect pleiotropic loci. mtCOJO is another multivariate method which uses summary results data but is designed to estimate genetic effects on a trait conditional on a correlated phenotype(s). Although MTAG and mtCOJO are not specifically designed to partition genetic effects into maternal and offspring components (i.e. by conditioning on a correlated genotype), they are user friendly and computationally efficient, and given the dearth of existing software packages to generate conditional genetic effect estimates using genome-wide summary results data, we were interested in investigating whether they would approximate the effects of interest accurately. Genomic SEM on the other hand is a highly flexible (albeit computationally intensive) method that allows users to specify a wide range of models to fit to the data. A summary of each of the methods and their underlying assumptions is provided in Table 1.”

- 3. Methods: A summary table giving the different methods, their assumptions, the data used, parameters used and the tests conducted for each method would be very helpful.**

Response: We have added a table summarizing each of the methods to the main text, which includes model assumptions, the data used and the genetic variants excluded from analyses (Table 1). Illustrations of each method can also be found in Supplementary Figures 1 through 4 and a detailed description of each method is contained in their legends to help orient the reader.

4. Methods: Why was the simulation under the null only carried out for mtCOJO?

Response: In our comparison of the methods, it appeared that mtCOJO had increased “power” to detect effects originating from both maternal and offspring GWAS- however in both cases the magnitude of the effect size was estimated incorrectly. We did not see this increase for any of the other methods tested and so were curious as to how the mtCOJO method performed under the null hypothesis of no genetic effect (i.e. whether it would also show inflation under the null). We realize however, that these additional analyses may be confusing to readers and go beyond the major remit of the manuscript (which was to determine which methods accurately estimate conditional maternal and offspring genetic effects on offspring phenotypes from summary results data). Given that mtCOJO does not provide accurate estimates of conditional maternal and offspring genetic effects (and therefore should not be used in this context), we have made the decision to exclude these simulations and associated text from an already very large manuscript.

5. Results: Again a summary table would be helpful.

Response: We have provided a summary across each of the methods for the computational time and inflation of the test statistics in Table 2. We have added to this table a summary of the comparison of the effect size estimates and standard errors between each of the methods and the SEM using individual level data.

6. Discussion: Please focus in the discussion on describing the strengths and limitations of this study, i.e., comparing 4 methods for genetic summary data to obtain SEM estimates against one individual SEM method addressing the same question. For example, are all the 4 summary methods comparable in concept? Are one or two comparisons sufficient to say one method is better than another? At the limit are all methods equivalent to the individual level SEM? Why does Genomic SEM do better than the other methods? Is it a general property of Genomic SEM or was it a chance finding in this particular example? Is Genomic SEM better than the linear approximation to SEM because it does not use the fixed constants given in the equations for the linear approximation to SEM?

Response: Despite being user friendly software packages, our results show conclusively that MTAG and mtCOJO are not suitable for generating accurate estimates of conditional parental and offspring genetic effects. In contrast, genomic SEM produced estimates and standard errors that were very similar to if the same analyses had been performed using individual level data. Although one can never be sure that genomic SEM will perform optimally in every situation and generate effect estimates and standard errors similar to had individual level data been analyzed, we believe that genomic SEM’s intrinsic flexibility and ability to model cryptic relatedness and sample overlap mean that it is likely to perform well more generally when estimating conditional parental and offspring genetic effects from summary results data. We have added the following paragraph to the discussion to address the reviewer’s concerns:

“There are several strengths and limitations of our study. Firstly, not all of the five methods we examined were developed to condition on a correlated genotype (i.e. parental and/or offspring genotype in the present context). In particular, MTAG and mtCOJO are multivariate

methods that were specifically developed for other purposes (i.e. to increase power to detect pleiotropic loci, and to estimate genetic effects conditional on a correlated phenotype respectively). Previous work has shown that both methods perform excellently when applied to the situations for which they were originally developed^{13,16}. However, given the paucity of existing software to estimate conditional effects from summary results data especially genome-wide, we were interested in whether these user-friendly software packages could also be used to approximate conditioning on a correlated genotype, and generate accurate parental and offspring specific genetic effects on a phenotype.

Of the comparisons that we made across all the different methods (i.e. comparing effect size estimates and standard errors, computational time, inflation in the test statistics, ability to account for sample overlap and ability to be extended to incorporate additional genetic effect estimates), genomic SEM performed best on all comparisons except computational time. In contrast, mtCOJO and MTAG did not yield accurate estimates or SEs of conditional maternal and/or offspring genetic effects. Although, we based our conclusions on findings from a single large dataset, we believe that our results are likely to hold more generally and are a reflection of genomic SEM's flexibility in being able to accurately model the relationship between parental and offspring genotypes (i.e. neither MTAG nor mtCOJO specifies this relationship accurately- see below for further discussion of this point) and genomic SEM's ability to take into account sample overlap and cryptic relatedness across the different GWAS (i.e. the weighted linear model doesn't estimate sample overlap and utilization of this information is not optimal in ordinary SEM)."

Reviewer #2:

- 7. In the methods section the authors refer to the standard errors of the SEM using covariance matrices as under estimated when there is no sample overlap, however in the subsequent sentence they state that this could be due to the presence of sample overlap not accounted for in the SEM using individual level data. These two sentences appear conflicting – it could be that the SEM using individual level data is over estimating the standard errors – can the authors clarify these sentences and justify why they describe the standard errors as under estimated?**

Response: We apologise for the confusion. We have clarified these sentences in the results section (new text is underlined): “This could be due to the small sample overlap that was estimated by LD score regression (8,396 individuals were estimated to overlap both GWAS when in reality there were no individuals overlapping. This could be due to e.g. LD score regression identifying cryptic relatedness across the GWAS) and included in the SEM using covariance matrices. We showed in the initial paper describing the SEM⁹ that there is an increase in power, due to a reduction in the standard error, when individuals with both their own and their offspring's phenotype are included in a model that specifies this relationship. Therefore, the 8,396 individuals estimated to overlap between the GWAS will result in a reduction of the standard error in the SEM using summary statistics (where we model this relationship) in comparison to the SEM using individual level data (where no sample overlap is modelled).”

- 8. They also describe the standard errors from mtCOJO as overestimated and underestimated when discussing the results of simulations – while the simulations suggest the standard errors are in fact overestimated under the null, could the authors justify why they describe the standard errors as underestimated for the 300 loci previously associated with birth weight rather than for example the possibility that conditioning on phenotype rather than genotype increases power at these loci.**

Response: Please see our response to Reviewer #1 point 4. We realize that these analyses may be confusing to readers and go beyond the major remit of the manuscript (which was to determine which methods accurately estimate conditional maternal and offspring genetic effects on offspring phenotypes from summary results data). Given that mtCOJO does not provide accurate estimates of conditional maternal and offspring genetic effects (and therefore should not be used in this context), we have made the decision to exclude these simulations and associated text from an already very large manuscript.

- 9. When examining the summary statistics across the genome the SEM using summary data finds a large number of genome wide significant loci compared to the other methods, many of which have not previously been associated with birth weight which are described as false positives. Given the inflation in the test statistics many of these are likely to be false positives, however the authors state that standard errors are “underestimated” for SNPs where maternal and offspring genetic effects are in opposite directions. These loci are less likely to be discovered in unadjusted GWAS of birth weight – it would help the reader to discuss how likely it is that discovery of SNPs with opposite maternal and fetal directions which may be masked in previous GWAS contribute to these genome-wide significant SNPs given the difference in sample size between previous GWAS of birth weight and the current analysis.**

Response: We have added the following explanation to the discussion:

“It is likely that the SNPs that reached genome-wide significance using this method, but are unknown birth weight associated loci, are false positives for three main reasons; 1) as seen in the Manhattan plots presented in the Supplementary Material, the majority of these SNPs are singletons and not part of LD blocks, 2) none of the other methods included in the comparison identified these loci and 3) the most recent GWAS of birth weight²⁴, which include the data in this study in addition to data from many birth cohorts and also partitioned the genetic effect into maternal and offspring components, did not identify these loci.”

- 10. In the discussion the authors suggest that MTAG is the most powerful method for detecting novel loci as its effect estimates are the sum of maternal and offspring effects at that locus. The increased power is only true when effects are in the same direction – there will be cases where maternal and offspring effects are in opposite directions and the sum of the effects is zero. In these cases MTAG would have reduced power to detect associations and the other methods described would have more power than MTAG to detect these loci.**

Response: We thank the reviewer for bringing this point up which has highlighted an inaccuracy in our description of the MTAG methodology. However, the reviewer’s contention that “increased power is only true when effects are in the same direction- there will be cases where maternal and offspring effects are in opposite directions and the sum of the effects is zero”- is not quite accurate either. The ability of MTAG to increase power for locus discovery depends strongly upon the genome-wide genetic correlation between variables (i.e. higher in magnitude the better) and the degree to which the pattern of effects at individual loci is consistent with the genome-wide correlation more broadly (i.e. the more consistent the better). We have changed the paragraph in the discussion to the following to reflect these issues:

“Given that MTAG is not designed to estimate maternal and offspring specific effects at individual loci, it was not surprising that it performed poorly in terms of accurately partitioning the genetic effect into maternal and offspring components. This is because the

MTAG model estimates combined pleiotropic genetic effects on both phenotypes (i.e. in the context of this manuscript, a pleiotropic effect on own birthweight and offspring birthweight c.f. **Supplementary Figure 2**). Intuitively, the MTAG model “borrows power” from a correlated phenotype to increase overall power to detect association. Previous work by other groups suggests that MTAG is likely to be a powerful approach if the goal of the investigator is locus discovery- particularly in situations where the magnitude of the genetic correlation between variables is high, and where the pattern of genetic effects at the individual SNP level is concordant with the genetic correlation between the phenotypes across the genome more broadly¹³. In contrast, our results suggest that if the focus is on locus characterization/accurately partitioning effects into maternal and offspring components (e.g. for downstream MR analyses where it is important to block potentially pleiotropic paths through related individuals (Evans et al 2019 *Int J Epidemiol*)) then one of the SEM based procedures discussed in this manuscript will be more appropriate.”

Reviewer #3:

11. Slightly more space should be devoted to explaining the principles of the Genomic SEM approach (lines 252 --)

Response: We have added the following text to the methods:

“Genomic SEM¹⁷ is a highly flexible, two stage multivariate statistical method for analysing the joint genetic architecture of traits using GWAS summary results statistics. In stage one, a K order genetic covariance matrix is estimated from the genome-wide summary results data of K GWAS using LD score regression¹⁷. Estimates of the standard errors for each of the variance-covariance terms which account for sample overlap between the GWAS are also obtained. This stage contrasts with “ordinary” structural equation modelling which uses a covariance matrix obtained from individual level data (e.g. a covariance matrix derived from individual level genotype, own birthweight and offspring birthweight). In stage two, a user specified model is then fit to the genetic covariance matrix in an attempt to explain the underlying pattern of genetic correlations across the traits in terms of a series of latent genetic variables. The model can be augmented through the addition of observed SNP variables, providing the opportunity to perform multivariate tests of association between individual SNPs and phenotypes, estimate the conditional effect of SNPs, and in some cases increase statistical power to detect association. In this manuscript, we create a path model based on standard biometrical genetics theory to model the genetic relationship between own and offspring birthweight, and use genomic SEM to estimate conditional maternal (paternal) and offspring specific genetic effects. The specific model that we fit to the birth weight data is depicted in **Supplementary Figure 4**.”

12. At lines 488--, before launching into a description of the computational strategy/procedure, some readers might welcome a description of the problem in scientific terms (defining the three effects components), and a brief discussion helping their intuition of how the problem is "translated" into a SEM model.

Response: We have added the following text to provide the readers with some intuition of the problem:

“Given an offspring’s genotype is correlated ~ 0.5 with both parental genotypes, and the offspring could influence parental decision to have additional children (for example, due to certain behavioural traits), it is important to adjust for offspring specific genetic effects when investigating the genetic determinants of fertility. We will refer to this offspring specific genetic effect as sibling specific effects as we are estimating it using the number of siblings an individual has. Additionally, we will estimate the female specific genetic effect on fertility using the number of children mothered and the male specific genetic effect on fertility using the number of children fathered.”

13. The Discussion should clearly distinguish which considerations are a direct output of the submitted paper, and which are reported from previous studies.

Response: We have gone through the discussion and added referencing and language to properly distinguish considerations reported from previous studies with those that are a direct output from the submitted manuscript.

Reviewers' Comments:

Reviewer #1:

Remarks to the Author:

Thank you very much indeed for such comprehensive consideration of the issues raised. The paper seems much clearer now. I have no further comments

Reviewer #2:

Remarks to the Author:

I thank the authors for addressing my concerns in the revised manuscript. I have no further comments.